# First Successful Treatment of Advanced Intrahepatic Cholangiocarcinoma with Tasurgratinib Following Regulatory Approval: A Case Report from Clinical Practice

**DOI:** 10.3390/ijms26125586

**Published:** 2025-06-11

**Authors:** Yuta Maruki, Chigusa Morizane, Mao Okada, Shota Harai, Yoshikuni Nagashio, Susumu Hijioka, Hideki Ueno, Takuji Okusaka

**Affiliations:** 1Department of Hepatobiliary and Pancreatic Oncology, National Cancer Center Hospital, 5-1-1 Tsukiji, Chuo-ku, Tokyo 104-0045, Japan; cmorizan@ncc.go.jp (C.M.); maookada@ncc.go.jp (M.O.); shhara@ncc.go.jp (S.H.); yonagash@ncc.go.jp (Y.N.); shijioka@ncc.go.jp (S.H.); hiueno@ncc.go.jp (H.U.); tokusaka@ncc.go.jp (T.O.); 2Department of Experimental Therapeutics, National Cancer Center Hospital, 5-1-1 Tsukiji, Chuo-ku, Tokyo 104-0045, Japan

**Keywords:** intrahepatic cholangiocarcinoma, *FGFR2* fusion, tasurgratinib, targeted therapy, case report, biliary tract cancer

## Abstract

Intrahepatic cholangiocarcinoma (iCCA) is a malignancy with limited treatment options in advanced stages. Recently, targeted therapies against fibroblast growth factor receptor 2 (*FGFR2*) fusions have emerged as a promising approach for selected patients. Tasurgratinib, a selective *FGFR1–3* inhibitor, was approved in Japan in 2024 for second-line treatment of *FGFR2* fusion-positive biliary tract cancer. We report the case of a 55-year-old female with advanced iCCA harboring an *FGFR2-BICC1* fusion, who experienced a rapid clinical response to tasurgratinib following disease progression on gemcitabine, cisplatin, and durvalumab (GCD). Following the failure of GCD therapy, treatment with oral tasurgratinib was initiated at 140 mg/day and subsequently reduced to 105 mg/day due to Grade 2 diarrhea. Within weeks, imaging and tumor markers indicated a partial response, accompanied by a reduction in ascites, and improved performance status. The response sustained for several months without evidence of disease progression. Notably, no substantial clinical hyperphosphatemia or anorexia was observed during treatment. This is the first report to describe the real-world clinical efficacy of tasurgratinib in an iCCA patient with *FGFR2-BICC1* fusion. Our findings suggest that tasurgratinib can provide a rapid and durable response with manageable toxicity in molecularly selected patients who have progressed on standard therapies.

## 1. Introduction

Surgical resection is the primary curative treatment for biliary tract cancer (BTC). However, chemotherapy as palliative treatment is advised for inoperable cases or postoperative recurrence. Gemcitabine plus cisplatin (GC) [1], gemcitabine plus S-1 [2], and gemcitabine plus cisplatin plus S-1 [3] are the recommended first-line treatments for BTC. Recently, two regimens have been shown to benefit from the addition of the immune checkpoint inhibitors (ICIs) durvalumab [4] and pembrolizumab [5]. However, the therapeutic efficacy of these medical treatments is insufficient. Recently, therapies targeting cancer driver gene abnormalities have been developed, because BTC is a relatively common cancer for which therapeutic target gene abnormalities have been identified [6].

Isocitrate dehydrogenase 1 (*IDH1*) and fibroblast growth factor receptor 2 (*FGFR2*), the two druggable genes, are frequently reported as therapeutic target gene abnormalities in intrahepatic cholangiocarcinoma (iCCA) [7]. After first-line treatment, two *FGFR2* inhibitors were used as second-line agents. The options for second-line treatment were expanded with the approval of pemigatinib [8] in March 2021 and futibatinib [9] in June 2023 for patients positive for the *FGFR2* fusion gene or rearrangement. Tasurgratinib (E7090, TASFYGO^®^, Eisai Co., Ltd., Tokyo, Japan) [10] was recently approved in Japan in September 2024 for the second-line treatment of unresectable *FGFR2* fusion or rearrangement-positive BTC. Tasurgratinib, an oral small-molecule selective inhibitor of *FGFR* 1, 2, and 3, was evaluated in an open-label, single-arm, multicenter, Phase II study in Japanese and Chinese patients [11].

Another unique feature of tasurgratinib is that break-apart fluorescence in situ hybridization (FISH) can be used as a companion diagnostic method for *FGFR2* gene fusion or rearrangement before the prescription of tasurgratinib [10]. Herein, we report our experience with a valuable clinical case involving tasurgratinib, where a significant response to tasurgratinib was observed in a patient in whom first-line treatment with gemcitabine + cisplatin + durvalumab (GCD) therapy failed.

## 2. Case Presentation

A 55-year-old female was diagnosed with a resectable intrahepatic cholangiocarcinoma (iCCA). Her medical history included a hepatitis B virus infection. She received a right hepatectomy, and this pathological stage was II (pathological T2N0M0, UICC TNM 8th edition). Postoperative adjuvant therapy was not administered. Surveillance for recurrence was conducted using contrast-enhanced computed tomography (CT) every three months during the first two years following surgery, and subsequently every six months from the third year onward. Three years after the radical resection, an isolated lung metastasis recurred and was surgically resected. One year later, another recurrence of lung metastasis occurred, and radiofrequency ablation was performed. Eight years after the liver resection, multiple masses were detected in the pelvis. Endoscopic ultrasound-guided tissue acquisition (EUS-TA) was performed via the rectum, leading to a diagnosis of adenocarcinoma. Immunohistochemical staining revealed positivity for N-cadherin and CRP and negativity for S100P, supporting the diagnosis of intrahepatic cholangiocarcinoma of the small duct type. As the lesions were deemed unresectable (Figure 1), palliative treatment with gemcitabine, cisplatin, and durvalumab (GCD) was initiated. After two cycles of GCD therapy, imaging revealed a slight increase in metastatic pelvic tumors, and the tumor marker CA19-9 level showed an upward trend (from 100 U/mL to 120 U/mL). Comprehensive genomic profiling (CGP) was performed. After four cycles of GCD therapy, the metastatic pelvic tumor showed further mild enlargement, accompanied by an increase in ascites. Treatment was discontinued due to disease progression. Based on the results of CGP, which revealed an *FGFR2-BICC1* fusion, oral tasurgratinib was initiated as second-line therapy.

Tasurgratinib administration was initiated at a dose of 140 mg/day. Two weeks after initiation of the treatment, Grade 2 diarrhea developed as an adverse event, leading to a three-day treatment interruption. After the interruption, the diarrhea improved, and treatment was resumed at a reduced dose of 105 mg/day. A therapeutic effect was observed on the 9th day after the start of oral administration, with a decrease in the tumor marker CA19-9 from 98 U/mL to 59 U/mL and a further decrease to 29 U/mL on the 28th day. Imaging (CT scan) evaluation conducted 47 days after treatment initiation showed a reduction in the size of the pelvic tumor, indicating a partial response (PR) to the therapy (Figure 2). Additionally, the ascites in the pelvic cavity decreased, and the patient’s body weight dropped from 50.9 kg at the start of treatment to 47 kg at the time of evaluation one month later. Following treatment initiation, tumor shrinkage continued for three months; however, Grade 2 fatigue developed, prompting a three-week treatment break. After the break, the tumor marker CA19-9 level had increased to 58 U/mL. As fatigue had improved during the break, treatment was resumed at a pre-break dose of 105 mg/day. Tasurglatinib treatment has been continuing for five months without disease progression, and therapy remains tolerable (Figure 3). Notably, common adverse events associated with *FGFR* inhibitors, such as hyperphosphatemia and anorexia, were not observed in this patient, and phosphate-lowering agents like lanthanum carbonate were not required.

## 3. Discussion

In recent years, chemotherapy for BTC has increasingly incorporated molecular targeted therapies and immune checkpoint inhibitors alongside conventional cytotoxic anticancer drugs. Different types of cancer express distinct target molecules, and BTC has a higher proportion of therapeutic target molecules compared to pancreatic and other cancers. Inda et al. [12] reported that 73% of Japanese patients with BTC had actionable gene mutations, with *KRAS* (26%) being the most common, followed by *TP53* (9%), *FGFR2* (8%), *CDKN2A/B* (7%), and *IDH1* (6%), although the rate of drug accessibility was only 16%.

In clinical trials of each *FGFR* inhibitor, time to response was reported as 2.7 months for pemigatinib, 2.5 months for futibatinib, and 1.87 months for tasurgratinib. Because the characteristics of each clinical study were slightly different, it was not possible to make a general judgment based on TTR levels alone. However, in our case, the tumor marker levels reduced approximately within two weeks after tasurgratinib administration. The reason for the early onset of the effect of tasurgratinib is not clear, but we look forward to seeing future case reports on the rapidity of the response. In patients with advanced *FGFR2* fusion-positive disease who require an early treatment response, tasurgratinib may also be considered a preferred option.

We speculate that there was a reason for the significant response to tasurgratinib in this case. An *FGFR2-BICC1* gene fusion was observed in our case. BICC1 is the most common fusion partner of FGFR2. [13]. *FGFR* fusions are classified as type I (non-receptor type, arising from N-terminal replacement by fusion partners, such as *BCR–FGFR1* and *ZMYM2–FGFR1*), type IIa (receptor type, also arising from N-terminal replacement by fusion partners; *FN1–FGFR1*, *KLK2–FGFR2*, and others), or type IIb (receptor type, with C-terminal replacement by fusion partners; *FGFR2–BICC1*, *FGFR3–TACC3*, and others) [14]. Zhang et al. reported that a patient harboring an *FGFR2-BICC1* gene fusion was treated with pemigatinib in combination with pembrolizumab plus systemic gemcitabine and oxaliplatin. After nine cycles of the combination therapy, the patient achieved a partial response [15]. In this case, durvalumab combination therapy was administered as first-line treatment, resulting in stable disease. A durable response is considered a characteristic feature of ICI therapy and may have contributed to the favorable outcome observed here [16]. Previously reported clinical trials were conducted when GC therapy was the standard first-line treatment, and the data reflects an era before the introduction of ICI therapy. Further investigation is warranted to evaluate the efficacy of FGFR inhibitors and the role of ICI therapy in BTC.

The companion diagnostic for tasurgratinib is fluorescence in situ hybridization (FISH) following Japanese approval. However, in our case, next-generation sequencing (NGS) technologies and comprehensive genomic profiling (CGP) tests (OncoGuide™ NCC Oncopanel System: NCC Oncopanel, Sysmex Corporation, Kobe, Japan) were used to detect *FGFR2* from biopsy specimens from lymph nodes. In a multicenter study in Japan (PRELUDE study) [17], we analyzed 423 patients with advanced BTC using the break-apart FISH method and found that 7.7% of iCCA and 4.8% of perihilar cholangiocarcinoma (PCC) cases were positive for the *FGFR2* fusion gene. Younger age (≤65 years) and HCV Ab- and/or HBs Ag-positivity were significantly associated with *FGFR2* rearrangements (logistic regression). It is also interesting to note that our patient was less than 65 years of age and HBs Ag-positive. Smoking, alcohol consumption, and hepatitis B virus infection have all been reported as potential risk factors associated with *FGFR2* fusion-positive iCCA [18].

The question for the future of BTC genetic testing is whether FISH or NGS testing should be offered first. NGS testing, which primarily relies on DNA analysis, has been reported to yield false-negative results when detecting *FGFR2* fusion genes. Cao et al. [19] reported a higher concordance rate between immunohistochemistry (IHC) and FISH (κ value = 0.778, *p* < 0.0001) than between IHC and NGS (κ value = 0.464, *p* = 0.072) in their comparison of FISH and NGS for iCCA, but more cases need to be accumulated in the future. The main advantage of FISH testing is that it can be performed with a small amount of tissue; however, its primary disadvantage is that it cannot detect abnormalities other than FGFR2 fusion genes. For iCCA, where many druggable genes can be identified, NGS testing—which can detect a broad range of genetic alterations—should be prioritized. However, in advanced iCCA or PCC, cases may arise where sufficient tissue cannot be obtained, making FISH testing an important alternative. Furthermore, since FISH testing has been reported to yield positive results in cases where NGS testing was negative, it is considered important to confirm negative NGS results with FISH testing.

## 4. Conclusions

Following failure of GCD therapy, tasurgratinib therapy demonstrated rapid and excellent efficacy of tasurgratinib.

## Figures and Tables

**Figure 1 ijms-26-05586-f001:**
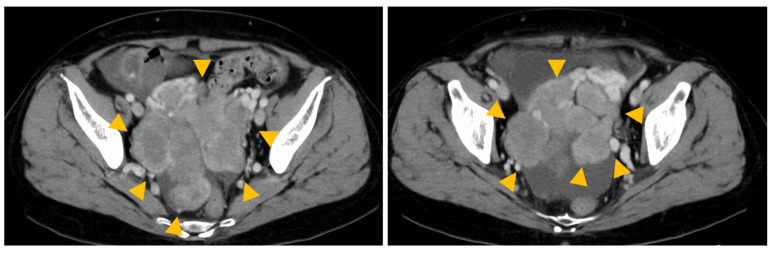
Images at the time of diagnosis of recurrence. The yellow arrow indicates tumors.

**Figure 2 ijms-26-05586-f002:**
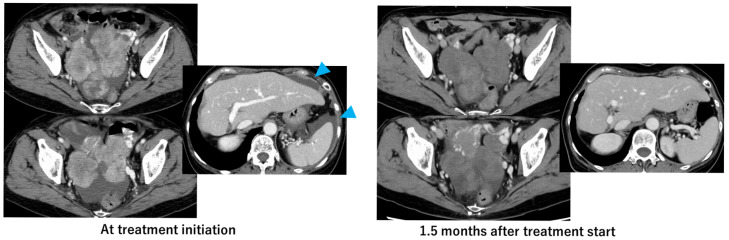
Tasurgratinib treatment effect. Images at treatment initiation and 1.5 months after starting treatment, showing marked reduction in ascites. The blue arrows indicate ascites.

**Figure 3 ijms-26-05586-f003:**
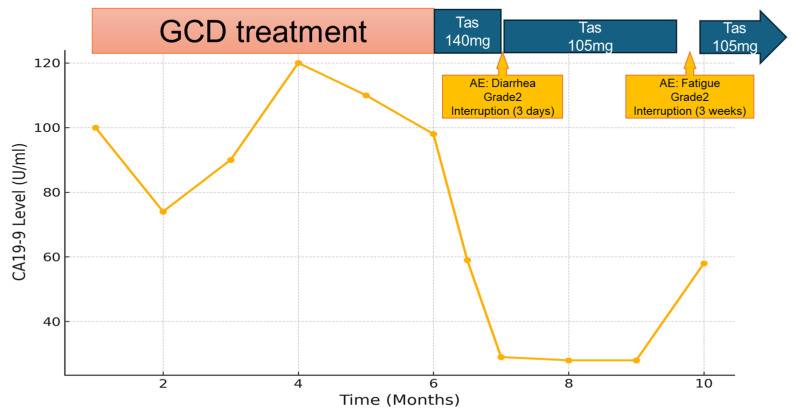
Course of medication treatment. GCD treatment followed by tasurgratinib (140 mg, then 105 mg). Adverse events included Grade 2 diarrhea (resulting in a 3-day interruption) and Grade 2 fatigue (resulting in a 3-week interruption).

## Data Availability

The original contributions presented in this study are included in the article. Further inquiries can be directed to the corresponding author(s).

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
