# Peer review of "First Successful Treatment of Advanced Intrahepatic Cholangiocarcinoma with Tasurgratinib Following Regulatory Approval: A Case Report from Clinical Practice"

_ijms, 2025, doi:10.3390/ijms26125586_

Round 1
Reviewer 1 Report
Comments and Suggestions for Authors
Your paper is a case report accompanied by a literature review on treatment of advanced intrahepatic cholangiocarcinoma with tasurgratinib. In light of the fact that recent studies have suggested that alcohol in relatively small amounts may lead to malignancies, your center in Japan should consider collaborating with other centers to increase the number of patients in the study.
Author Response
Reviewer1
Your paper is a case report accompanied by a literature review on treatment of advanced intrahepatic cholangiocarcinoma with tasurgratinib. In light of the fact that recent studies have suggested that alcohol in relatively small amounts may lead to malignancies, your center in Japan should consider collaborating with other centers to increase the number of patients in the study.
⇒Thank you for this valuable comment. A recent study involving Japanese patients has reported an association between a history of alcohol consumption and FGFR2 fusion-positive biliary tract cancer. In particular, a multicenter collaborative study in Japan would be beneficial to increase patient numbers and allow for a more robust analysis of lifestyle-related risk factors such as alcohol intake. We have now added a statement reflecting this point in the Discussion section.
“Smoking, alcohol consumption, and hepatitis B virus infection have all been reported as potential risk factors associated with FGFR2 fusion-positive iCCA.” This sentence has been added to the Discussion section at line 154.

Reviewer 2 Report
Comments and Suggestions for Authors
I commend The authors on a well written case report. My only suggestion is a little bit more history about the patient, and if the patient was on regular surveillance for recurrence after the initial surgery. Perhaps the author should consider adding a little bit more detail details about her follow and If the patient was lost to follow up. Another suggestion would be to add in discussion. What is the general surveillance program for similar malignancy.
Author Response
Reviewer2
I commend The authors on a well written case report. My only suggestion is a little bit more history about the patient, and if the patient was on regular surveillance for recurrence after the initial surgery. Perhaps the author should consider adding a little bit more detail details about her follow and If the patient was lost to follow up. Another suggestion would be to add in discussion. What is the general surveillance program for similar malignancy.
⇒Thank you very much for your positive feedback and constructive suggestions.
We agree with the importance of providing additional clinical background on the patient. Accordingly, we have added further details regarding her surveillance history following the initial surgery, including information on whether she was under regular follow-up or had been lost to follow-up.
We believe these additions enhance the clarity and clinical relevance of the case, and we greatly appreciate your insightful recommendations.
“Postoperative adjuvant therapy was not administered. Surveillance for recurrence was conducted using contrast-enhanced computed tomography (CT) every three months during the first two years following surgery, and subsequently every six months from the third year onward.” This sentence has been added to the Case Report at line 64.

Reviewer 3 Report
Comments and Suggestions for Authors
This is a well-written case report. I have just two minor comments for the authors to address to further improve the clarity of the manuscript.
- I am not entirely sure the authors can claim this as the first report of the real-world case. There are already previously reported cases presented at conferences. The authors should clarify their statement to accurately reflect the novelty of their report.
- I would suggest that the authors include arrows or markers in the CT images to clearly indicate the lesions, This would enhance the clarity and help readers better interpret the findings.
Author Response
Reviewer3
This is a well-written case report. I have just two minor comments for the authors to address to further improve the clarity of the manuscript.
I am not entirely sure the authors can claim this as the first report of the real-world case. There are already previously reported cases presented at conferences. The authors should clarify their statement to accurately reflect the novelty of their report.
⇒Thank you very much for your valuable and important comment. As this case report represents the first to be published following the regulatory approval of tasurgratinib, we have revised the title as below.
“First Successful Treatment of Advanced Intrahepatic Cholangiocarcinoma with Tasurgratinib Following Regulatory Approval: A Case Report from Clinical Practice”
I would suggest that the authors include arrows or markers in the CT images to clearly indicate the lesions, This would enhance the clarity and help readers better interpret the findings.
⇒Thank you very much. Arrows in the CT images indicate the location of the lesion.
In addition, we have revised Affiliation 2 to: Department of Experimental Therapeutics, National Cancer Center Hospital, Tokyo, Japan.
